# Ultrasound Imaging of the Facial Muscles and Relevance with Botulinum Toxin Injections: A Pictorial Essay and Narrative Review

**DOI:** 10.3390/toxins14020101

**Published:** 2022-01-27

**Authors:** Wei-Ting Wu, Ke-Vin Chang, Hsiang-Chi Chang, Lan-Rong Chen, Chen-Hsiang Kuan, Jung-Ting Kao, Ling-Ying Wei, Yunn-Jy Chen, Der-Sheng Han, Levent Özçakar

**Affiliations:** 1Department of Physical Medicine and Rehabilitation, National Taiwan University Hospital, Bei-Hu Branch, Taipei 10845, Taiwan; wwtaustin@yahoo.com.tw (W.-T.W.); lchen@livemail.tw (L.-R.C.); dshan1121@yahoo.com (D.-S.H.); 2Department of Physical Medicine and Rehabilitation, College of Medicine, National Taiwan University, Taipei 10048, Taiwan; 3Center for Regional Anesthesia and Pain Medicine, Wang-Fang Hospital, Taipei Medical University, Taipei 11600, Taiwan; 4Department of Physical Medicine and Rehabilitation, Taichung Veterans General Hospital, Taichung 407219, Taiwan; s19801041@gm.ym.edu.tw; 5Division of Plastic Surgery, Department of Surgery, National Taiwan University Hospital, Taipei 10048, Taiwan; chkuan0408@gmail.com; 6Department of Dermatology, National Taiwan University Hospital, Bei-Hu Branch, Taipei 10845, Taiwan; jungtingkao@yahoo.com.tw; 7Department of Dentistry, National Taiwan University Hospital, Bei-Hu Branch, Taipei 10845, Taiwan; f98422011@ntu.edu.tw; 8Department of Dentistry, National Taiwan University Hospital, Taipei 10048, Taiwan; chenyj@ntu.edu.tw; 9Department of Physical and Rehabilitation Medicine, Medical School, Hacettepe University, Ankara 06100, Turkey; lozcakar@yahoo.com

**Keywords:** ultrasonography, cosmetic, face, injection, rejuvenation

## Abstract

High-resolution ultrasound is preferred as the first-line imaging modality for evaluation of superficial soft tissues, such as the facial muscles. In contrast to magnetic resonance imaging and computed tomography, which require specifically designated planes (axial, coronal and sagittal) for imaging, the ultrasound transducer can be navigated based on the alignment of facial muscles. Botulinum toxin injections are widely used in facial cosmetic procedures in recent times. Ultrasonography is recognized as a useful tool for pre-procedure localization of target muscles. In this pictorial review, we discuss the detailed sonoanatomy of facial muscles and their clinical relevance, particularly with regard to botulinum toxin injections. Furthermore, we have summarized the findings of clinical studies that report ultrasonographic imaging of facial muscles.

## 1. Introduction

High-resolution ultrasound (US) has emerged as one of the most convenient imaging tools for evaluation of superficial soft tissues [1,2]. Most facial muscles are superficially located and are clearly visualized using US. Furthermore, in contrast to magnetic resonance imaging and computed tomography, which usually require specifically designated planes (axial, coronal, and sagittal) for accurate imaging, the US transducer can be positioned/navigated based on the alignment of facial muscles. Although several studies have reported visualization of facial muscles [3,4,5], a systematic US scanning protocol is warranted to guide clinicians in routine practice. 

Botulinum toxin injections are widely used in recent times for cosmetic dermatologic procedures involving the face [6,7]. US imaging theoretically facilitates pre-procedure localization of the target muscles and prevents injury to vital neural/vascular structures. The use of US imaging in guiding the injection of botulinum toxin carries certain benefits for the treatment of dystonia, spasticity, hemifacial spasms and re-innervation synkinesis, including improvement in therapeutic efficacy and reduction of adverse effects compared with the landmark guided approach [8]. In this pictorial review, we discuss in detail the sonoanatomy of the facial muscles and their clinical relevance, particularly with regard to botulinum toxin injections. Furthermore, we have summarized the findings of clinical studies that report US imaging of facial muscles. All US images presented in this article were obtained using a 10–25 MHz high-frequency linear transducer (X-Cube 90, Alpinion Medical Systems Co. Ltd., Anyang, Korea).

## 2. Overview of the Superficial Facial Anatomy

In contrast to other body parts, the subcutaneous layers of the face appear well organized. The superficial musculo-aponeurotic system (SMAS) refers to a fibrous network of inelastic tissues located deep within the subcutaneous tissues, with occasional investment into the underlying muscular layer [9,10]. SMAS is observed in the forehead, temporal, parotid, zygomatic, buccal, infraorbital, and mental regions. The facial nerve is closely associated with the SMAS; the proximal branches of the facial nerve, mainly the temporal, zygomatic, and marginal mandibular branches, course deep to the SMAS. In contrast, the sensory (ophthalmic and maxillary) branches of the trigeminal nerve run superficial to the SMAS [10]. A face-lift rejuvenation procedure essentially involves SMAS tightening. Figure 1 shows the facial muscles described in this article.

## 3. Muscles of the Upper Face

### 3.1. Frontalis 

#### 3.1.1. Anatomy

The frontalis muscle originates from the galea aponeurotica (a layer of dense connective tissue that extends over the cranium) and is inserted into the orbicularis oculi muscle. It is innervated by the temporal branch of the facial nerve and receives its blood supply from the supraorbital and supratrochlear arteries. Contraction of the frontalis muscle raises the eyebrows and wrinkles the forehead [11].

#### 3.1.2. Scanning Technique

The transducer is initially placed in the horizontal plane at one fingerbreadth cranial to the eyebrow. The frontalis muscle covers the frontal bone and is visible beneath the SMAS (Figure 2A) [5]. The transducer is subsequently placed in the sagittal plane to observe the frontalis muscle in its long axis (Figure 2B); the muscle is visible gliding against the frontal bone as the subject elevates the eyebrows.

#### 3.1.3. Clinical Relevance

Aging is associated with the development of wrinkles over the forehead, perpendicular to the course of the frontalis muscle. Botulinum toxin injections relax the frontalis muscle and are therefore useful to minimize wrinkles [12]. It is recommended that botulinum toxin be injected 2 cm cranial to the eyebrow to avoid inadvertent paralysis of the levator palpebrae superioris and subsequent ptosis. Bell’s palsy is associated with complete paralysis of the frontalis muscle, which results in flattening of skin over the forehead and drooping of the eyebrow on the affected side [4].

### 3.2. Temporalis

#### 3.2.1. Anatomy

The temporalis muscle originates from the parietal bone of the skull and the superior temporal surface of the sphenoid bone and is inserted on the coronoid process of the mandible and retromolar fossa. It is innervated by the anterior division of the mandibular nerve and receives its blood supply from the deep temporal artery. Contraction of the temporalis muscle elevates and retracts the mandible [13].

#### 3.2.2. Scanning Technique

The transducer is initially placed along the zygomatic arch in the horizontal plane and is subsequently moved cranially; the temporalis muscle is visualized lying in the temporal fossa (Figure 3A) [1]. The transducer is also rotated 90° in the coronal plane to visualize the muscle in its long axis (Figure 3B); its distal portion is present beneath the zygomatic arch (invisible portion) and the upper masseter muscle (visible part of the muscle).

#### 3.2.3. Clinical Relevance

Myofascial trigger points inside the temporalis muscle can lead to tension headaches. In 2003, McGuigan et al. [14] reported a case of recalcitrant temporal and frontal headache in a patient in whom computed tomography revealed bilateral diffuse swelling and nodular thickening in the temporalis muscle. Botulinum toxin injection into the temporalis muscle relieves temporomandibular joint pain. Age-induced temporalis muscle atrophy results in temporal hollowing. Hyaluronic acid fillers can be administered between the subgaleal fascia and temporalis muscle to attain a youthful appearance [15].

### 3.3. Procerus

#### 3.3.1. Anatomy

The procerus muscle originates from the fascia over the lower portion of the nasal bone and is inserted into the skin overlying the lower forehead between the eyebrows. It is innervated by the temporal branch of the facial nerve and receives its blood supply from the facial artery. Contraction of the procerus muscle depresses the medial end of the eyebrow and wrinkles the glabellar skin [16].

#### 3.3.2. Scanning Technique

The transducer is placed in the horizontal plane on the lower portion of the forehead between the eyebrows and slightly cranial to the nasion (also referred to as the nasal bridge) (Figure 4A). The short axis of the procerus can be visualized over the frontal bone and beneath the SMAS [3]. Pivoting the transducer in the sagittal plane facilitates visualization of its long axis along the nasal bone (Figure 4B).

#### 3.3.3. Clinical Relevance

Patients with progressive supranuclear palsy may present focal dystonia of the procerus muscle (referred to as the procerus sign) along with reduced blinking, lid retraction, and gaze palsy [17]. Botulinum toxin injections into the procerus muscle for cosmetic purposes eliminate age-induced horizontal furrow lines observed in the mid lower forehead [18].

### 3.4. Depressor Supercilii

#### 3.4.1. Anatomy

The depressor supercilii muscle originates from the medial orbital rim and is inserted on the medial wall of the bony orbit. It is innervated by the facial nerve and receives its blood supply from the supratrochlear artery. Contraction of this muscle leads to downward movement of the eyebrow [19].

#### 3.4.2. Scanning Technique

The transducer is placed in the horizontal plane on the middle third of the eyebrow. The depressor supercilii is observed lateral to the procerus muscle beneath the SMAS (Figure 5A). Slight rotation of the transducer toward the sagittal oblique plane enables visualization of the muscle in its long axis (Figure 5B).

#### 3.4.3. Clinical Relevance

The depressor supercilii was previously considered an extension/branch of the orbicularis oculi or corrugator supercilii muscle; however, it was subsequently confirmed to be a distinct muscle [19]. The depressor supercilii contributes to formation of oblique glabellar frown lines and can be inactivated by botulinum toxin injections for aesthetic purposes [18].

### 3.5. Corrugator Supercilii

#### 3.5.1. Anatomy

The corrugator supercilii originates from the supraorbital ridge and is inserted into the skin over the forehead, near the eyebrow. It is innervated by the facial nerve and receives its blood supply from the ophthalmic artery. Contraction of this muscle pulls the eyebrow downward and medially [20].

#### 3.5.2. Scanning Technique

The center of the transducer is placed on the middle third of the eyebrow in the horizontal plane. The corrugator supercilii is visualized deep to the depressor supercilii, with its lateral border beneath the orbicularis oculi (Figure 6A). The supraorbital foramen serves as a landmark located immediately below the corrugator supercilii. Slightly pivoting the medial edge of the transducer toward the medial orbital rim facilitates visualization of this muscle in its long axis [21] (Figure 6B).

#### 3.5.3. Clinical Relevance

The corrugator supercilii is referred to as the “frowning muscle”; its contraction results in the formation of vertical wrinkles on the forehead. Botulinum toxin injections into the corrugator supercilii effectively flatten the glabellar region and normalize the contour of the medial eyebrow in patients with thyroid eye diseases [22].

### 3.6. Orbicularis Oculi

#### 3.6.1. Anatomy

The orbicularis oculi muscle originates from the frontal bone, medial palpebral ligament, and lacrimal bone and is inserted on the lateral palpebral raphe. It is innervated by the temporal and zygomatic branches of the facial nerve and receives its blood supply from the ophthalmic, zygomatico-orbital, and angular arteries. Contraction of this muscles closes the eyelid [23].

#### 3.6.2. Scanning Technique

The transducer is placed over the eyebrow in the horizontal plane, and the muscle is visualized lateral to the corrugator supercilii (Figure 7A). The transducer can be relocated more laterally to observe the orbicularis oculi overlying the temporoparietal fascia [5] (Figure 7B). Table 1 summarizes the scanning techniques used for all the aforementioned muscles of the upper face.

#### 3.6.3. Clinical Relevance

The orbicularis oculi plays an important role in the blink reflex, which is used to evaluate the integrity of the trigeminal and facial nerves. The muscle often serves as the target for upper eyelid blepharoplasty, which is used in the treatment of sunken eyes [23].

## 4. Muscles of the Middle Face

### 4.1. Nasalis

#### 4.1.1. Anatomy

The nasalis originates from the maxilla and is inserted into the nasal bone. It is innervated by the buccal branch of the facial nerve and receives its blood supply from the superior labial artery. Contraction of the muscle leads to compression of the nasal bridge and depression of the nasal tip [24].

#### 4.1.2. Scanning Technique

The transducer is placed in the oblique coronal plane along the nasal cartilage; the nasalis is visualized in its short axis above the cartilage. The transducer can be redirected to the oblique horizontal plane to observe the muscle along its long axis (Figure 8) [25].

#### 4.1.3. Clinical Relevance 

Contraction of the nasalis enlarges the nose and stretches the nostril. Overactivation of the muscle produces bunny lines (diagonal lines that radiate downward from either side of the nose); botulinum toxin injections soften and erase these lines [18].

### 4.2. Levator Labii Superioris Alaeque Nasi

#### 4.2.1. Anatomy

The levator labii superioris alaeque nasi originates from the nasal bone and is inserted into the nostril and upper lip. It is innervated by the buccal branch of the facial nerve and receives its blood supply from the angular branch of the facial and infraorbital branches of the maxillary arteries. Contraction of the muscle elevates the upper lip to expose the upper teeth [26].

#### 4.2.2. Scanning Technique

The transducer is placed in an oblique horizontal plane that passes through the nasal crease. The muscle is observed lateral to the nasalis and medial to the levator labii superioris (Figure 9A). The transducer can be rotated to the oblique sagittal plane to observe the muscle along its long axis (Figure 9B). The angular branch of the facial artery serves as an important anatomical landmark; the artery courses above the levator labii superioris alaeque nasi [25].

#### 4.2.3. Clinical Relevance

Overactivation of the levator labii superioris alaeque nasi results in a gummy smile, which is characterized by excessive gingival display on smiling. Botulinum toxin injection into this muscle can lengthen the upper lip to increase coverage of the gingiva and can also soften and minimize a prominent nasolabial fold [27].

### 4.3. Levator Labii Superioris

#### 4.3.1. Anatomy

The levator labii superioris originates from the medial infraorbital region and is inserted into the skin and muscle over the upper lip. It is innervated by the buccal branch of the facial nerve and receives its blood supply from the facial artery. Contraction of this muscle elevates the upper lip [28].

#### 4.3.2. Scanning Technique

The transducer is placed in the middle of the inferior orbital rim in the horizontal plane, which initially facilitates visualization of the orbicularis oculi, followed by visualization of the levator labii superioris in its short axis, which lies above the infraorbital foramen (Figure 10A). Following movement of the transducer in a more inferior direction, the levator labii superioris appears to be placed more medially (Figure 10B). The transducer can be rotated 90° to visualize the muscle in its long axis, as it courses over the maxilla [25] (Figure 10C).

#### 4.3.3. Clinical Relevance

Similar to the levator superioris alaeque nasi, the levator labii superioris is targeted to treat a gummy smile [29].

### 4.4. Levator Anguli Oris 

#### 4.4.1. Anatomy

The levator anguli oris originates from the maxilla and is inserted into the modiolus. It receives its blood supply from the facial artery and is innervated by the buccal branch of the facial nerve. Its contraction elevates the angle of the mouth [30].

#### 4.4.2. Scanning Technique

The scanning method is the same as that used for the levator labii superioris [25] which courses above the levator anguli oris (Figure 10B). The transducer can be rotated 90° to visualize the muscle in its long axis as it courses above the maxilla (Figure 10C).

#### 4.4.3. Clinical Relevance

The levator anguli oris is used for reconstruction of nasal defects secondary to surgical removal of tumors in this area. The muscle can also be considered as a target for botulinum toxin injections to correct a gummy smile [6].

### 4.5. Zygomaticus Minor

#### 4.5.1. Anatomy

The zygomaticus minor originates from the zygomatic bone and is inserted on the skin of the upper lip. It is innervated by the buccal branch of the facial nerve and receives its blood supply from the facial artery. Contraction of this muscle elevates the upper lip [31].

#### 4.5.2. Scanning Technique

The transducer is placed over the lateral inferior corner of the orbital rim in the horizontal plane. The origin of the zygomaticus minor can be visualized beneath the orbicularis oculi muscle (Figure 11A). The medial end of the transducer can be redirected toward the lateral half of the upper lip to visualize the muscle along its long axis (Figure 11B) [25].

#### 4.5.3. Clinical Relevance

The levator labii superioris is partially covered by the levator labii superioris alaeque nasi and the zygomaticus minor. Botulinum toxin injections into the zygomaticus minor are usually considered to treat a gummy smile or facial asymmetry in patients with excessive upward and lateral displacement of the upper lip [31].

### 4.6. Zygomaticus Major

#### 4.6.1. Anatomy

The zygomaticus major originates from the lateral aspect of the zygomatic bone and is inserted into the modiolus (a small fibromuscular structure) of the mouth. It is innervated by the buccal and zygomatic branches of the facial nerve and receives its blood supply from the superior labial branch of the facial artery. This muscle participates in elevation and contraction of the angle of the mouth [32].

#### 4.6.2. Scanning Technique

The transducer is placed over the inferior lateral edge of the orbital rim in the horizontal plane. The zygomaticus major is usually visualized lateral to the zygomaticus minor (Figure 12A); however, its fibers may blend with the zygomaticus minor, and the two muscles may be indistinguishable. The medial end of the transducer can be redirected toward the angle of the mouth to visualize the muscle in its long axis (Figure 12B) [33].

#### 4.6.3. Clinical Relevance

Infiltration of the botulinum toxin into the zygomaticus major during injection of the adjacent muscles may cause partial lip ptosis. Botulinum toxin injections into the zygomaticus major can be considered for correction of facial asymmetry in patients in whom the angle of the mouth is excessively drawn backward/upward when smiling [18].

### 4.7. Masseter

#### 4.7.1. Anatomy

The masseter muscle has two heads (superficial and deep). The superficial head originates from the anterior two-thirds and the deep head from the posterior third of the zygomatic arch. The masseter muscle is inserted on the lateral surface of the mandibular ramus and angle. It is innervated by the masseteric branch of the mandibular nerve and receives its blood supply from the masseteric artery. Contraction of the masseter causes mandibular elevation and protrusion [1].

#### 4.7.2. Scanning Technique

The transducer is placed along the zygomatic arch in the horizontal plane and is subsequently moved caudally to visualize the masseter muscle in its short axis (Figure 13A). The transducer can be redirected 90° to visualize the masseter bridge between the zygomatic arch and the mandible (Figure 13B) [1]. Table 2 summarizes the scanning techniques used for all the aforementioned muscles of the middle face.

#### 4.7.3. Clinical Relevance

Masseter hypertrophy can occur secondary to bruxism or habitual tooth grinding, which may lead to a square face (widening of the lower third of the face). Botulinum toxin injections into the mandibular insertion of the masseter muscle are useful to manage teeth grinding and jaw contouring, although caution is warranted to avoid injury to the parotid gland [34].

## 5. Muscles of the Lower Face

### 5.1. Orbicularis Oris

#### 5.1.1. Anatomy

The orbicularis oris originates from the medial aspect of the maxilla and mandible, the perioral skin/muscles, as well as the modiolus, and is inserted into the skin and mucosa of the lip. It is innervated by the buccal branch of the facial nerve and receives its blood supply from the facial, maxillary, and superficial temporal arteries. Contraction of this muscle leads to compression of the mouth and protrusion of the lip [35].

#### 5.1.2. Scanning Technique

The transducer is placed over the lower philtrum (a vertical groove between the nose and upper lip) and labiomandibular crease in the horizontal plane. The orbicularis oris appears as a thin hypoechoic band between the two layers of connective tissue (Figure 14A) [36]. The muscle can also be visualized by placing the transducer in the horizontal plane just inferior to the lower lip (Figure 14B).

#### 5.1.3. Clinical Relevance

Botulinum toxin injections into the orbicularis oris can be considered to decrease perioral vertical rhytids. A small volume of the toxin should be injected into the superficial portion of the muscle to avoid impairment of phonation and sucking functions [37].

### 5.2. Buccinator

#### 5.2.1. Anatomy

The buccinator muscle originates from the alveolar process of the maxilla, the buccinator ridge of the mandible, and the pterygomandibular raphe. It is inserted onto the modiolus, and its fibers blend with those of the orbicularis oris. It is innervated by the buccal branch of the facial nerve and receives its blood supply from the buccal artery. Its contraction compresses the cheek against the molar teeth, which facilitates whistling [38].

#### 5.2.2. Scanning Technique

The transducer is placed in the horizontal plane between the zygomatic arch and the mandible to initially visualize the masseter in its short axis [1]. The transducer is subsequently relocated in a more anterior direction, and the buccinator muscle is visualized in its long axis between the undersurface of the masseter muscle and the buccopharyngeal fascia (Figure 15).

#### 5.2.3. Clinical Relevance

Facial synkinesis, defined as inappropriate and inadvertent movements of the facial muscles during certain voluntary facial expressions, is a common sequela of facial nerve palsy. The buccinator muscle is commonly involved in facial synkinesis, which can be treated by botulinum toxin injections [39].

### 5.3. Risorius

#### 5.3.1. Anatomy

The risorius muscle originates from the parotid fascia and buccal skin and is inserted on the modiolus. It is innervated by the buccal branch of the facial nerve and receives its blood supply from the superior labial branch of the facial artery. Its contraction extends the angle of the mouth laterally [40].

#### 5.3.2. Scanning Technique

The transducer is initially placed in the horizontal plane to visualize the masseter muscle in its short axis [1]. The transducer is subsequently relocated in a more anterior direction to view the muscle as it emerges from the superficial fascia of the superficial head of the masseter (Figure 15).

#### 5.3.3. Clinical Relevance

Accidental infiltration of botulinum toxin into the risorius may occur during injection into the masseter muscle. Caution is warranted to avoid facial asymmetry.

### 5.4. Mentalis

#### 5.4.1. Anatomy

The mentalis muscle originates from the anterior mandible and is inserted into the chin. It is innervated by the mandibular branch of the facial nerve and receives its blood supply from the inferior labial branch of the facial artery and the mental branch of the maxillary artery. Contraction of the mentalis elevates the chin and results in lower lip protrusion [41].

#### 5.4.2. Scanning Technique

The transducer is placed over the midline of the chin in the horizontal plane. The mentalis muscle is identified in its short axis above the mandible (Figure 16A). The transducer can be redirected to the sagittal plane to visualize the muscle in its long axis (Figure 16B) [5].

#### 5.4.3. Clinical Relevance

Hereditary geniospasm, a rare movement disorder, is characterized by episodic involuntary movements of the mentalis muscle [42]. Mentalis overactivity may cause blunting of the contour or increased horizontal wrinkles over the chin, and botulinum toxin injections can be considered in such cases.

### 5.5. Depressor Labii Inferioris 

#### 5.5.1. Anatomy

The depressor labii inferioris originates from the oblique line of the mandible and is inserted into the integument of the lower lip [43]. It is innervated by the mandibular branch of the facial nerve and receives its blood supply from the inferior labial branch of the facial artery and the mental branch of the maxillary artery. Contraction of the muscle depresses the lower lip inferolaterally.

#### 5.5.2. Scanning Technique

The transducer is initially placed over the midline of the chin to visualize the mentalis muscle and is subsequently relocated more laterally to identify the depressor labii inferioris above the lateral edge of the mentalis (Figure 17A) [3]. The transducer can be redirected 90° to view the muscle in its long axis (Figure 17B).

#### 5.5.3. Clinical Relevance

Overactivation of the depressor labii inferioris may cause a droopy appearance of the face secondary to lowering of the lower lip; intramuscular botulinum toxin injections are useful in such cases [18].

### 5.6. Depressor Anguli Oris

#### 5.6.1. Anatomy

The depressor anguli oris originates from the oblique line of the mandible and is inserted into the modiolus [43]. It is innervated by the mandibular branch of the facial nerve and receives its blood supply from the facial artery. Contraction of the muscle leads to depression of the angle of the mouth.

#### 5.6.2. Scanning Technique

The transducer is placed over the midline of the chin and is subsequently moved laterally. The mentalis muscle is visualized, followed by the depressor labii inferioris and depressor anguli oris (Figure 17A). The transducer can be redirected to identify the muscle in its long axis as it courses over the mandible (Figure 17B) [44]. Table 3 summarizes the scanning techniques used for all the aforementioned muscles of the lower face.

#### 5.6.3. Clinical Relevance

Overactivity of the depressor anguli oris lowers the mouth commissure, which leads to the expression of sadness or frustration. Botulinum toxin injections elevate the angle of the mouth and improve an individual’s smile [18].

## 6. Literature Review

### 6.1. Literature Search 

Although the present article is a pictorial essay and narrative review, we performed a systematic literature search of the PubMed, Medline, and Web of Science databases (without language limitations) from inception to January 2022 to identify articles relevant to US imaging of facial muscles. The following keywords and their combinations were used: ultrasound, sonography, ultrasonography and face, facial muscles. The following search strategy had been used for literature search: (“ultrasound” or “ultrasonography” or “sonography”) AND (“face” or “facial muscle”). The inclusion criteria were cross-sectional, case-control, cohort and randomized controlled studies that use ultrasound imaging to visualize the group of facial muscles. The exclusion criteria comprised (1) nonhuman studies, (2) case report or series, (3) review articles and (4) studies that only investigate a single muscle. Our review only included articles that described more than one facial muscle.

### 6.2. Results

The process of the literature search was detailed in the Appendix A (Appendix A: flow chart of literature search; Search Results from Different Databases). Seven articles were included in this study. In 2013, Alfen et al. [3] used US to investigate the facial muscles in 12 healthy adults and in one patient with myotonic dystrophy. The authors observed fair and excellent reproducibility of muscle thickness measurements for the orbicularis oris, the procerus and the levator labii superioris muscles, respectively. The echogenicity in most scanned facial muscles was higher in the patient with myotonic dystrophy than in the control group. In 2013, Volk et al. [44] investigated US imaging of the bilateral facial muscles in 40 adults and observed that men showed thicker depressor anguli oris and thinner orbicularis oculi muscles than women. Furthermore, women showed a significant side-to-side difference in the orbicularis oculi.

In 2014, Volk et al. [36] used US imaging to investigate bilateral facial muscles in 140 volunteers aged between 21 and 93 years. Nearly all muscles (except the temporalis) were symmetrical in size. The body mass index was significantly correlated with the size of most target muscles. Volk et al. [4] used a similar scanning protocol to investigate facial muscles in 20 patients with chronic facial palsy and observed a significantly smaller muscle size on the paralyzed side in the orbicularis oculi, orbicularis oris, depressor labii inferioris, depressor anguli oris, and mentalis muscles. However, no side-to-side asymmetry was identified in the chewing muscles (temporalis and masseter).

In 2016, Volk et al. [45] investigated the correlation between US and electromyographic findings in 44 patients with unilateral peripheral facial palsy and observed that facial muscle thickness at rest and during activation was best correlated with insertional activity on electromyography. In contrast, changes in muscle thickness between the resting state and during contraction were significantly associated with voluntary activity on electromyography. The aforementioned correlations were significant 14 days after onset of palsy.

In 2019, Abe et al. [5] investigated the reliability of US imaging to measure facial muscle thickness. The intra-rater reliability expressed using intra-class correlation coefficients ranged from 0.425 (for the orbicularis oris) to 0.943 (for the frontalis). The minimal important difference varied from 0.25 mm (for the orbicularis oculi) to 1.82 mm (for the masseter).

In 2021, Hormazabal-Peralta et al. [25] investigated the depth of the mid-face muscles and the distribution of vessels using high-resolution US in 88 volunteers. The authors clearly identified the facial artery, facial vein, angular artery, angular vein, and perforator vessels of the mid face. Additionally, a significant sex-based difference was observed in the depth of the orbicularis oculi, levator labii superioris alaeque nasi, and zygomaticus minor muscles.

We had several viewpoints regarding sonoanatomy described in the seven included articles. First, the scanning techniques for the large-sized facial muscles (such as the masseter and temporalis) and adjacent bony landmarks were consistent across the enrolled studies. There were some variations in the imaging methods for small-sized muscles such as the levator labii superioris and zygomaticus minor. Second, most of the enrolled studies used the short-axis views to examine the facial muscles, whereas their longitudinal fiber arrangement could not be clearly depicted. In our protocol, we employed the short- and long-axis views to visualize the target muscles, which could compensate the weakness of the methods reported in the previous studies. Third, during the literature search, we did not identify studies comparing US imaging with other imaging tools (like magnetic resonance imaging). A prospective trial would be needed to investigate the validity of US imaging for assessing the texture of the facial muscles in comparison with other imaging tools.

## 7. Future Perspectives and Limitations

Based on a review of the included articles, we observed that US may serve as a useful tool to quantify facial muscle thickness, as well as to evaluate muscle echotexture. Variations in the reliability of measurements across different muscles may be attributable to changes in scanning methods, which highlights the importance of a standardized and stepwise evaluation protocol (as described in this article). Although the botulinum toxin is widely used for rejuvenation and in cosmetic dermatology, few studies have investigated the role of US guidance in comparison with landmark-based injections. The facial muscles are thin and superficially located; therefore, in our opinion, the use of high-frequency (hockey-stick) transducers and an out-of-plane injection technique may be useful. Future studies are warranted to conclusively establish the clinical effectiveness and safety of such interventions.

Several limitations of using US guidance for injecting facial muscles should be acknowledged. First, utilization of US guidance for most facial muscles may be excessive because the majority of facial muscles can be identified with anatomical surface landmarks, such as most movement disorders (hemifacial spasm, blepharospasm and other forms of facial dystonia) [46]. Second, the localization of the target muscles by US imaging is challenging to validate. The cosmetologists may consider using US imaging while injecting muscles besides some vital neurovascular structures, such as the facial artery and facial nerves. A guideline made by the consensus of the cosmetologists is needed in the future to determine in what circumstance would the use of US imaging be necessary because most cosmetologists do not utilize the technique currently.

## 8. Conclusions

High-resolution US enables the delineation of facial muscles, and any asymmetry and targets for possible interventions can be promptly evaluated. Botulinum toxin injections are usually performed based on surface anatomy, and currently, US guidance is rarely used for this purpose. Further prospective studies are warranted to establish the feasibility and advantages of US imaging and guidance in the management of facial (muscle) disorders. Clinicians should consider the complementary role of US and electrodiagnostic tests in these patients. Notably, US guidance can facilitate accurate needle insertion during electromyography of the aforementioned thin/small muscles.

## Figures and Tables

**Figure 1 toxins-14-00101-f001:**
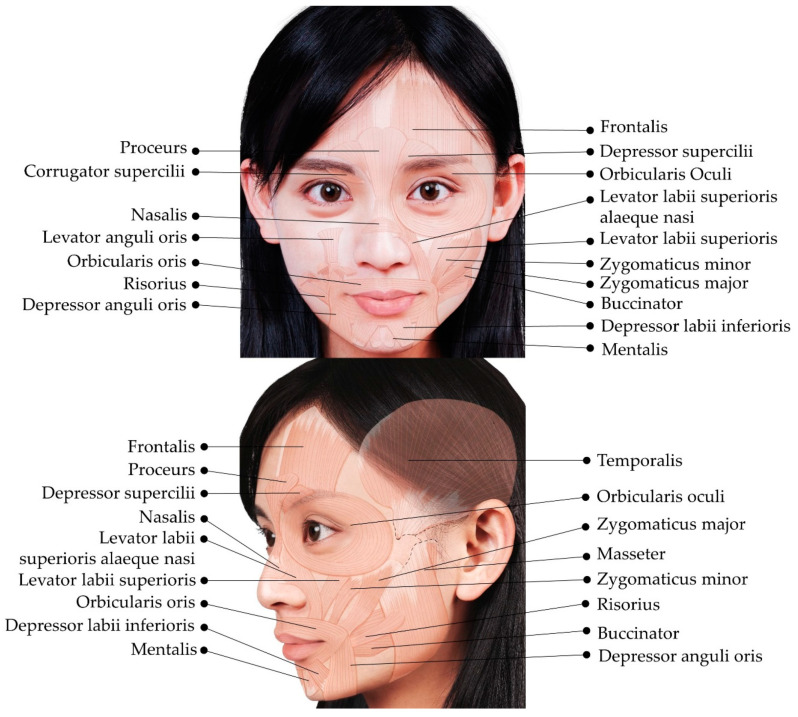
Illustration showing the facial muscles. The images represent an adaptation of our co-author’s (H.-C.C) face with permission for publication.

**Figure 2 toxins-14-00101-f002:**
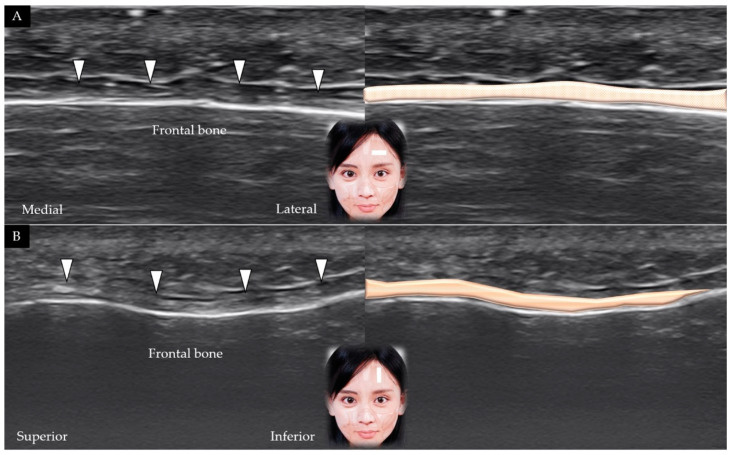
Ultrasound scans and schematic representation of the frontalis muscle (white arrowheads and brown shade) in its short (**A**) and long (**B**) axis.

**Figure 3 toxins-14-00101-f003:**
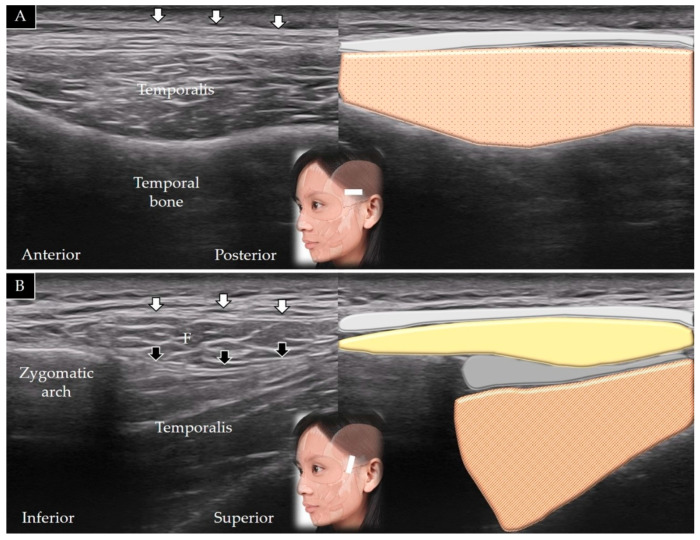
Ultrasound scans and schematic representation of the temporalis muscle in its short (**A**) and long (**B**) axis. White arrows and light grey shade, temporoparietal fascia; black arrows and dark grey shade, deep temporal fascia; F and yellow shade, fat; brown shade, temporalis muscle.

**Figure 4 toxins-14-00101-f004:**
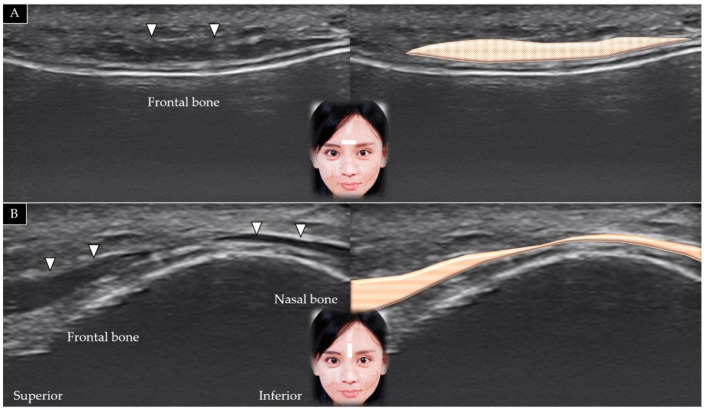
Ultrasound scans and schematic representation of the procerus muscle (white arrowheads and brown shade) in its short (**A**) and long (**B**) axis.

**Figure 5 toxins-14-00101-f005:**
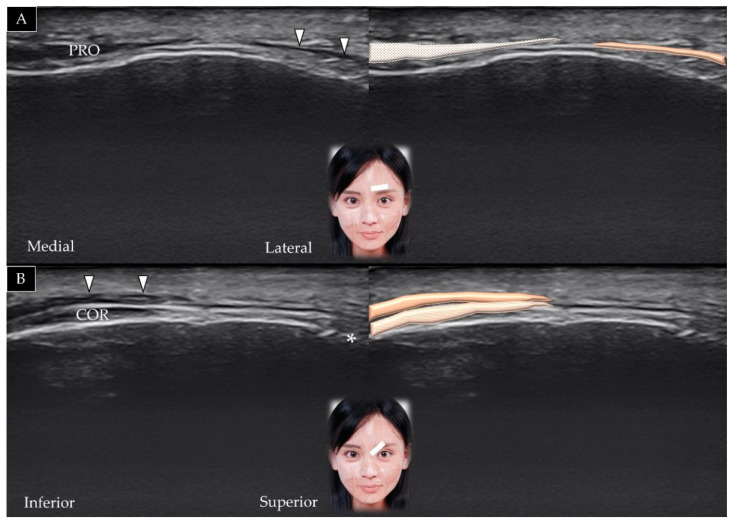
Ultrasound scans and schematic representation of the depressor supercilii (white arrowheads and brown shade) in its short (**A**) and long axis (**B**). *, supraorbital foramen; PRO and light grey shade, procerus; COR and light brown shade, corrugator supercilii.

**Figure 6 toxins-14-00101-f006:**
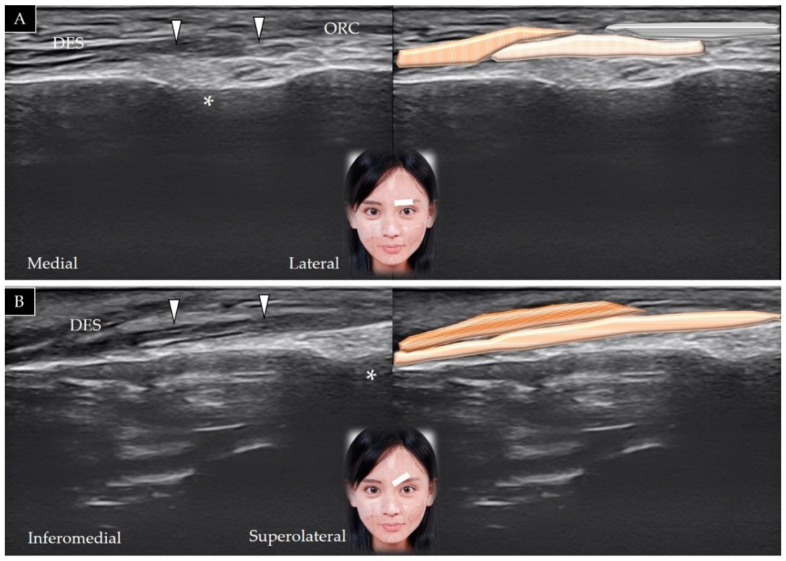
Ultrasound scans and schematic representation of the corrugator supercilii (white arrowheads and light brown shade) in its short (**A**) and long (**B**) axis. *, supraorbital foramen; DES and brown shade, depressor supercilii; ORC and grey shade, orbicularis oculi.

**Figure 7 toxins-14-00101-f007:**
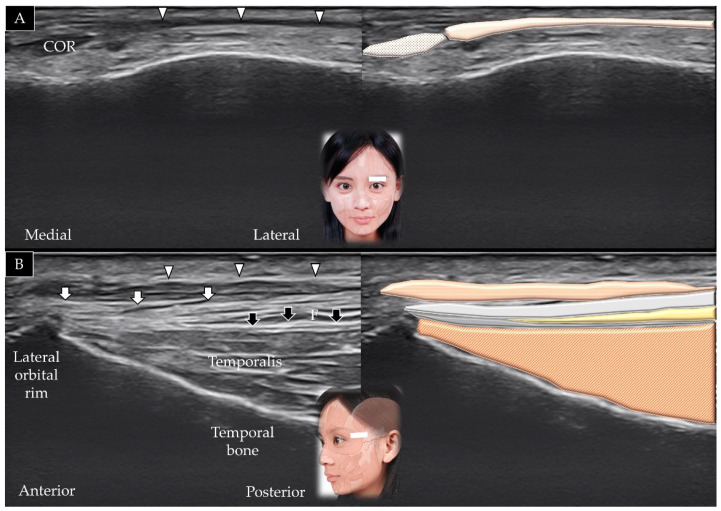
Ultrasound scans and schematic representation of the orbicularis oculi (white arrowheads and light brown shade) on top of the eyebrow (**A**) and lateral to the lateral orbital rim (**B**). White arrows and light grey shade, temporoparietal fascia; black arrows and dark grey shade, deep temporal fascia; COR and white dotted shade, corrugator supercilii; F and light yellow shade, fat; brown shade, temporalis muscle.

**Figure 8 toxins-14-00101-f008:**
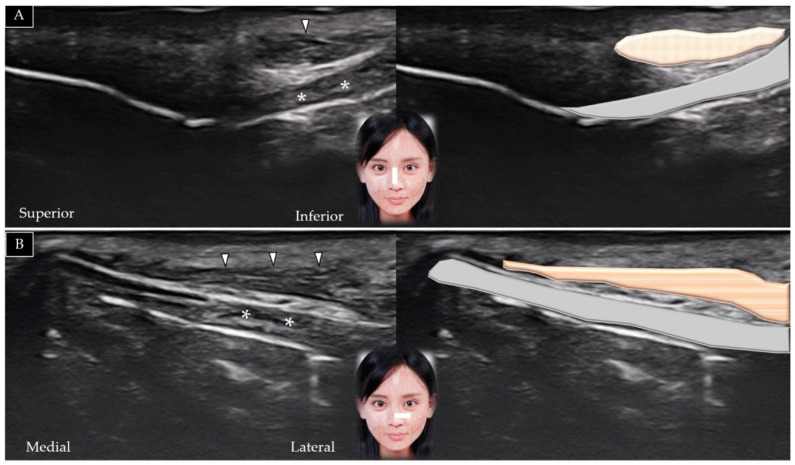
Ultrasound scans and schematic representation of the nasalis (white arrowhead and brown shade) in its short (**A**) and long (**B**) axis. * and grey shade, nasal cartilage.

**Figure 9 toxins-14-00101-f009:**
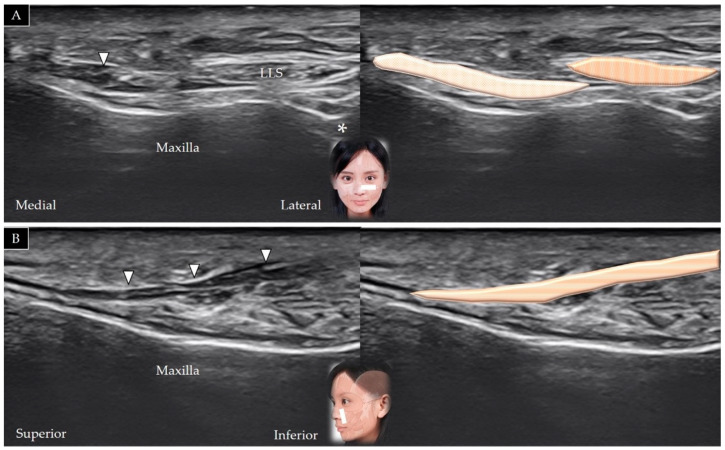
Ultrasound scans and schematic representation of the levator labii superioris alaeque nasi (white arrowheads and light brown shade) in its short (**A**) and long (**B**) axis. *, infraorbital foramen; LLS and brown shade, levator labii superioris.

**Figure 10 toxins-14-00101-f010:**
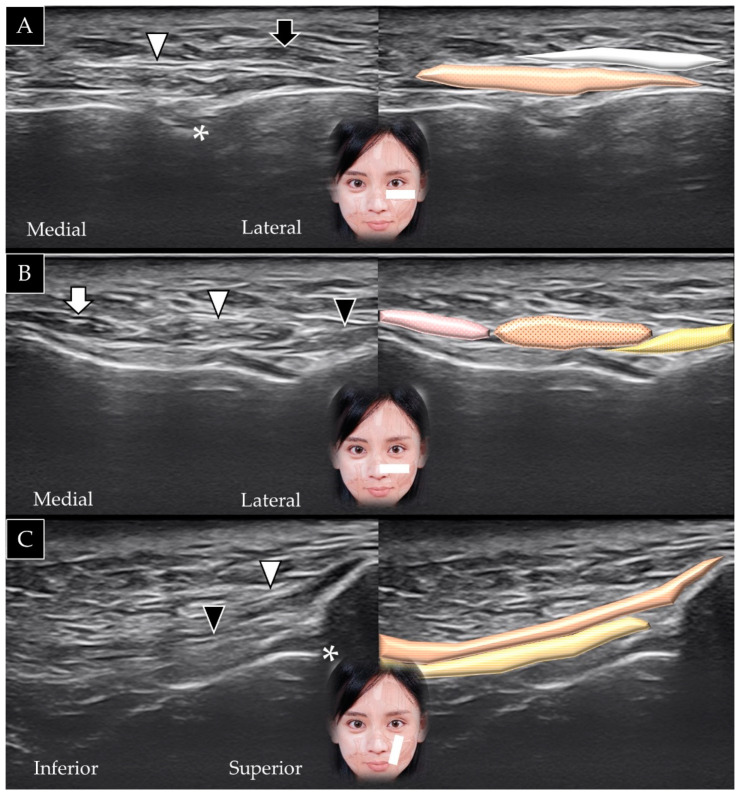
Ultrasound scans and schematic representation of the levator labii superioris (white arrowheads and brown shade) and the levator anguli oris (black arrowheads and yellow shade) in their short (**A**,**B**) and long (**C**) axes. White arrow and pink shade, levator labii superioris alaeque nasi; black arrow and white shade, orbicularis oculi; *, infraorbital foramen.

**Figure 11 toxins-14-00101-f011:**
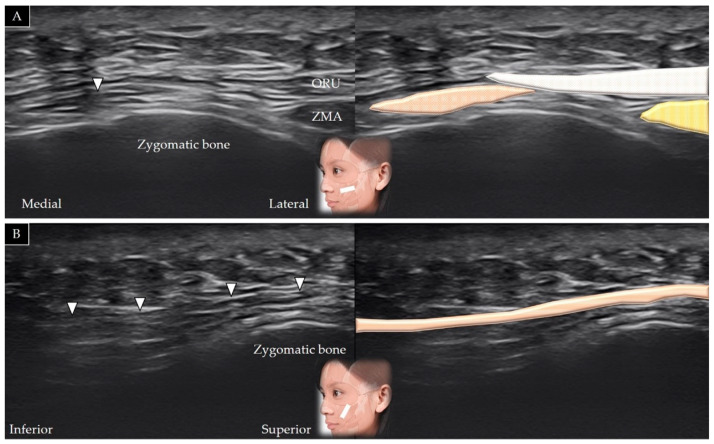
Ultrasound scans and schematic representation of the zygomaticus minor (white arrowheads and brown shade) in its short (**A**) and long (**B**) axis. ORU and white shade, orbicularis oculi; ZMA and yellow shade, zygomaticus major.

**Figure 12 toxins-14-00101-f012:**
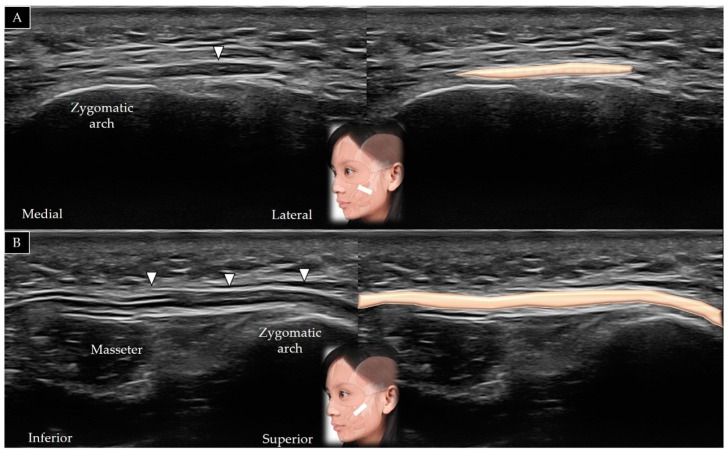
Ultrasound scans and schematic representation of the zygomaticus major (white arrowhead and brown shade) in its short (**A**) and long (**B**) axis.

**Figure 13 toxins-14-00101-f013:**
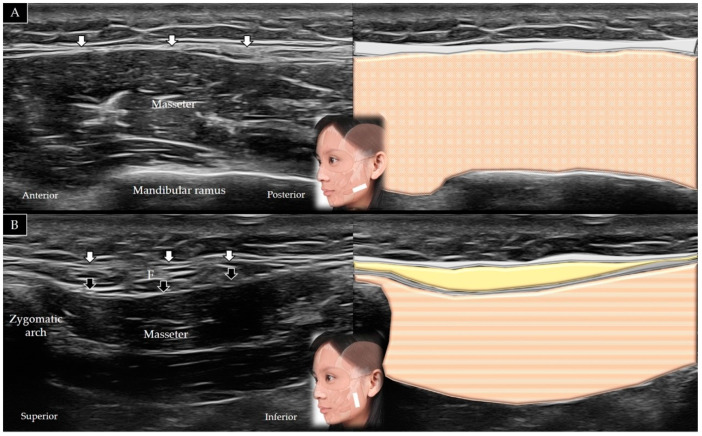
Ultrasound scans and schematic representation of the masseter in its short (**A**) and long (**B**) axis. White arrows and grey shade, superficial musculo-aponeurotic system; black arrows and dark grey shade, parotid-masseteric fascia; F and yellow shade, fat; brown shade, masseter muscle.

**Figure 14 toxins-14-00101-f014:**
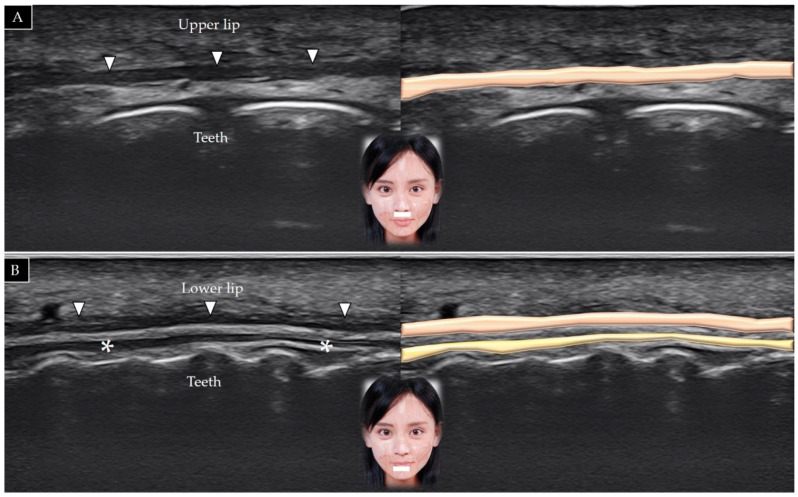
Ultrasound scans and schematic representation of the orbicularis oris (white arrowheads and brown shade) in its long axis over the lower philtrum (**A**) and over the labiomandibular crease (**B**). * and yellow shade, internal lining of the labial mucosa.

**Figure 15 toxins-14-00101-f015:**
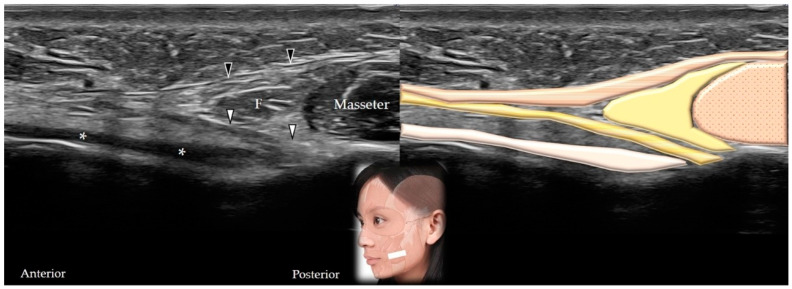
Ultrasound scan showing the buccinator (white arrowheads and dark yellow shade) and the risorius (black arrowheads and brown shade) (long-axis view). * and white shade, buccopharyngeal fascia; F and yellow shade, fat; dotted brown shade, masseter muscle.

**Figure 16 toxins-14-00101-f016:**
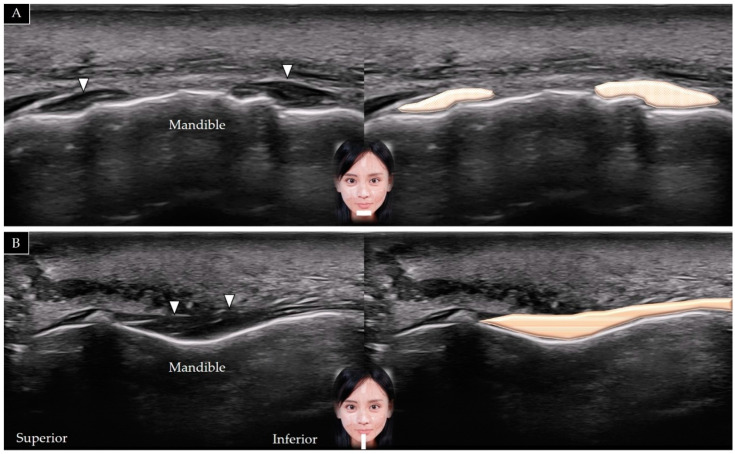
Ultrasound scans showing the mentalis (white arrowheads and brown shade) in its short (**A**) and long (**B**) axis.

**Figure 17 toxins-14-00101-f017:**
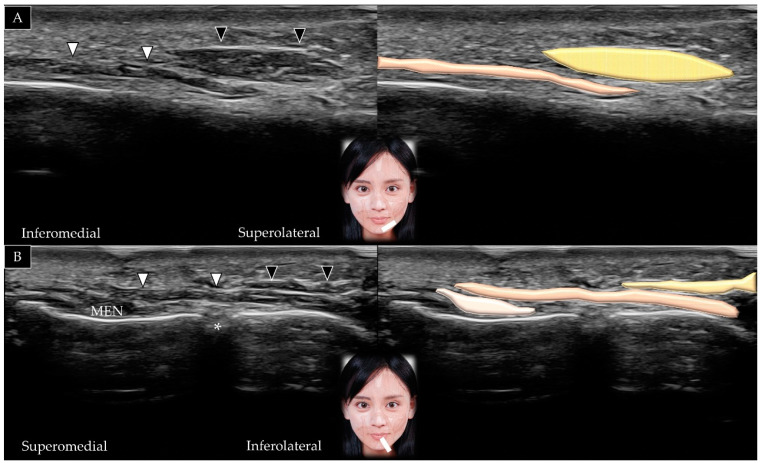
Ultrasound scans showing the depressor labii inferioris (white arrowhead and brown shade) and depressor anguli oris (black arrowhead and yellow shade) muscles in their short (**A**) and long (**B**) axes above the lateral edge of the mentalis. *, mental foramen; MEN and white shade, mentalis.

**Table 1 toxins-14-00101-t001:** Anatomy and scanning of the upper face muscles.

Muscle	Origin	Insertion	Transducer Position
Frontalis	Galea aponeurotica	Orbicularis oculi	In the horizontal plane at one fingerbreadth cranial to the eyebrow (short axis view)
Temporalis	Parietal and sphenoid bones	Coronoid process of the mandible and retromolar fossa	Along the zygomatic arch in the horizontal plane, then moved cranially (short axis view)
Procerus	Fascia over the lower portion of the nasal bone	Lower forehead between bilateral eyebrows	In the horizontal plane on the lower portion of the forehead between the eyebrows
Depressor supercilii	Medial orbital rim	Medial wall of the bony orbit	In the horizontal plane on the middle one third of the eyebrow
Corrugator supercilii	Supraorbital ridge	Skin of the forehead near the eyebrow	In the horizontal plane on the middle one third of the eyebrow
Orbicularis oculi	Frontal bone, medial palpebral ligament and lacrimal bone	Lateral palpebral raphe	In the horizontal plane over the lateral orbital wall (short/oblique axis view)

**Table 2 toxins-14-00101-t002:** Anatomy and scanning of the middle face muscles.

Muscle	Origin	Insertion	Transducer Position
Nasalis	Maxilla	Nasal bone	In the oblique coronal plane along the nasal cartilage (short axis view)
Levator labii superioris alaeque nasi	Nasal bone	Nostril and upper lip	In the oblique horizontal plane passing the nasal crease (lateral to the nasalis)
Levator labii superioris	Medial infraorbital region	Upper lip	In the middle of the inferior orbital rim in the horizontal plane (short axis view)
Levator anguli oris	Maxilla	Modiolus	Same as the overlying levator labii superioris
Zygomaticus minor	Zygomatic bone	Upper lip	In the horizontal plane over the lateral inferior corner of the orbital rim (underneath the orbicularis oculi)
Zygomaticus major	Zygomatic bone	Upper lip	In the horizontal plane over the inferior lateral edge of the orbital rim (lateral to the zygomatic minor)
Masseter	Zygomatic arch	Mandibular ramus and angle	In the horizontal plane along the zygomatic arch, then moved caudally (short axis view)

**Table 3 toxins-14-00101-t003:** Anatomy and scanning of the lower face muscles.

Muscle	Origin	Insertion	Transducer Position
Orbicularis oris	Maxilla and mandible, perioral skin and muscles and modiolus	Skin and mucosa of the lip	In the horizontal plane over the lower philtrum and labiomandibular crease
Buccinator	Maxilla, buccinator ridge of mandible and pterygomandibular raphe	Modiolus	In the horizontal plane between the zygomatic arch and mandible (emerging under the masseter)
Risorius	Parotid fascia and buccal skin	Modiolus	In the horizontal plane (emerging from the masseter superficial fascia)
Mentalis	Anterior mandible	Chin	Over the midline of the chin (short axis view)
Depressor labii inferioris	Oblique line of the mandible	Lower lip	Over the midline of the chin (lateral to the mentalis)
Depressor anguli oris	Oblique line of the mandible	Modiolus	Over the midline of the chin (lateral to the depressor labii inferioris)

## Data Availability

Data are contained within the main text and Appendix A of the present manuscript.

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
