# Peer review of "Ultrasound Imaging of the Facial Muscles and Relevance with Botulinum Toxin Injections: A Pictorial Essay and Narrative Review"

_toxins, 2022, doi:10.3390/toxins14020101_

Round 1
Reviewer 1 Report
This paper aimed to analyze and discuss the relevance of using ultrasound imaging as a guidance for botulinum toxin injections of the facial muscles. The opinion of this reviewer is that the manuscript is adequate for publication if they consider some minor recommendations.
Introduction: The authors provide information about US and botulinum toxin injections, but there are no information about the importance of using US when applying botulinum toxin injections. Please add it.
The information about the measures and the procedures is very well written and organized, congratulations.
On the description of the literature review (6.1, lines 462-467), please provide de search equations for each database, showing the descriptors used in each one. Moreover, in order to increase the reproducibility of the systematic search, please provide information about the terms (MeSH or free terms), the Boolean operators used (AND, OR) the inclusion and exclusion criteria of the studies. A flow chart summarizing all this information is highly recommended.
6.2 results: please consider to add in this section the type of study for each investigation included (crossover design, clinical trials…)
In the abstract (line 28) the authors state “in this pictorial review, we discuss in detail the somatoanatomy...” however, there is no discussion section in the manuscript. Please consider to add it in order to increase the quality of the paper, at the end of the results section and before the future perspectives. It could be interesting to compare the results obtained with similar investigations and to provide a perspective about the measures with US and other methods. Moreover, the discussion section will increase the consistence of the conclusions of the study.
Author Response
Reviewer 1
Comment:
This paper aimed to analyze and discuss the relevance of using ultrasound imaging as a guidance for botulinum toxin injections of the facial muscles. The opinion of this reviewer is that the manuscript is adequate for publication if they consider some minor recommendations.
Response:
We appreciate the kind comment from the reviewer which significantly improve the quality of the present manuscript. The manuscript has been revised according to the reviewer’s comment.
Comment:
Introduction: The authors provide information about US and botulinum toxin injections, but there is no information about the importance of using US when applying botulinum toxin injections. Please add it.
Response:
The use of ultrasound imaging in guiding the injection of botulinum toxin carries certain benefits for the treatment of dystonia, spasticity, hemifacial spasms and re-innervation synkinesis, including improvement in therapeutic efficacy and reduction of adverse effects compared with the landmark guided approach. The benefits have been highlighted by a review titled “Ultrasound-guided botulinum toxin injections in neurology: technique, indications and future perspectives” (Expert Rev Neurother 2014). The statement has been added in the introduction according to the reviewer’s comment. Please kindly refer to line 27-31.
Comment:
The information about the measures and the procedures is very well written and organized, congratulations.
Response:
We appreciate the positive comments from the reviewer.
Comment:
On the description of the literature review (6.1, lines 462-467), please provide the search equations for each database, showing the descriptors used in each one. Moreover, in order to increase the reproducibility of the systematic search, please provide information about the terms (MeSH or free terms), the Boolean operators used (AND, OR) the inclusion and exclusion criteria of the studies. A flow chart summarizing all this information is highly recommended.
Response:
We appreciate the kind comments from the reviewer. The following search strategy has been used for literature search: (“ultrasound” or “ultrasonography” or “sonography”) AND (“face” or “facial muscle”). The inclusion criteria were cross-sectional, case-control, cohort and randomized controlled studies that use ultrasound imaging to visualize the group of facial muscles. The exclusion criteria comprised (1) non-human studies, (2) case report or series, (3) review articles and (4) studies that only investigate a single muscle. Please kindly refer to line 476-482 in the revised manuscript. A flow chart has been added in the supplemental material to demonstrate the search process. Please kindly refer to Figure 1S.
Comment:
6.2 results: please consider to add in this section the type of study for each investigation included (crossover design, clinical trials…)
Response:
We appreciate the kind comments from the reviewer. The study types included for review encompassed cross-sectional, case-control, cohort and randomized controlled studies. We excluded case report and case series. Please kindly refer to the earlier response to the comment.
Comment:
In the abstract (line 28) the authors state “in this pictorial review, we discuss in detail the somatoanatomy...” however, there is no discussion section in the manuscript. Please consider to add it in order to increase the quality of the paper, at the end of the results section and before the future perspectives. It could be interesting to compare the results obtained with similar investigations and to provide a perspective about the measures with US and other methods. Moreover, the discussion section will increase the consistence of the conclusions of the study.
Response:
  We appreciate the kind comment from the reviewer. Through reading the seven articles, we would like to provide several viewpoints regarding sonoanatomy. First, the scanning techniques for the large-sized facial muscles (such as the masseter and temporalis) and adjacent bony landmarks were consistent across the included studies. There were some variations for small-sized muscles such as the levator labii superioris and zygomaticus minor.
Second, most of the included studies used the short-axis views to examine the facial muscles, whereas their longitudinal fiber arrangement could not be clearly depicted. In our protocol, we employed the short- and long-axis views to visualize the target muscles, which could compensate the weakness of the methods described reported in previous studies. Third, during the literature search, we did not identify studies comparing US imaging with other imaging tools (like magnetic resonance imaging). A prospective trial would be needed to investigate the validity of US imaging for assessing the texture of the facial muscles in comparison with other imaging tools. The following description has been added at the end of the results section and before the future perspectives according to the reviewer’s comments. Please kindly refer to line 521-534.

Reviewer 2 Report
The article summarized ultrasound to view facial muscles. As well anatomical contents that are helpful for the clinician is well described. These information would well be adapted to botulinum toxin injections which are widely used in facial cosmetic procedures in recent time.
The figures should be dramatically be changed. Please have the figures change in high resolution images. The muscular structure that are noted are hard to recognize, which muscle is which. The authors should change the figures in form of reader friendly. For example in the figure 6 I do not understand the yellow and orange shaded regions are representing.
Please fix all the figures that are simple but easy to recognize.
Overall, the article seems to fit the special issue in present form.
Author Response
Reviewer 2
The article summarized ultrasound to view facial muscles. As well anatomical contents that are helpful for the clinician is well described. The information would well be adapted to botulinum toxin injections which are widely used in facial cosmetic procedures in recent time.
The figures should be dramatically changed. Please have the figures change in high resolution images. The muscular structure that are noted are hard to recognize, which muscle is which. The authors should change the figures in form of reader friendly. For example, in the figure 6 I do not understand the yellow and orange shaded regions are representing.
Please fix all the figures that are simple but easy to recognize.
Overall, the article seems to fit the special issue in present form.
Response:
We appreciate the kind comments from the reviewer to improve the quality of the present study.
First, we increase the resolution of the figures to 600 dpi. However, as all the figures have been embedded in the word file, the resolution would be compromised. Furthermore, the facial muscle is very small and thin. We have tried our best to obtain the images of the best quality. We also checked the images in the included seven studies and clearly identified our image quality
was comparable to theirs. We have compiled all the figures in a file and submitted with the main text to the editorial group if post-processing is needed. We are grateful for the reviewer’s kind understanding.
Second, we also explained what the color shade represented in the figure legends of the revised manuscript. We believed it would significantly improve the clarity of the figures.
Reviewer 3 Report
Dear authors,
The paper is interesting and well-illustrated. However, I have a few comments as follows:
Figure 1 contains incorrect information. As the first figure of this manuscript delivers key anatomical information, I suggest that the Figure 1 be revised and the nomenclature be correctly used when indicating the facial muscles. For example, zygomatic major and minor should be revised as zygomaticus major and minor muscles.
What was indicated as the buccinator is actually the risorius muscle. The location of the temporalis muscle is also incorrect. Please carefully recheck the nomenclature and the location of the all the facial muscles before further submission. The superficial aponeurosis of the masseter muscle should be drawn longer than how it is depicted in the manuscript. Please refer to previous anatomical research papers that describe the morphology of the masseter muscle.
The temporal fossa indicated in Figure 3A should be revised. The temporalis muscle should be found in the temporal fossa as a hypoechoic structure in the ultrasonography image. However, the authors did not indicate the temporalis muscle in the temporal fossa. When the temporal fossa is observed in the ultrasonography image, the temporal bone surface can be found deep to the temporalis muscle as a hyperechoic structure.
The location of the transducer, ultrasonography image and corresponding illustration are incorrect in Figure 3. As the ultrasonography image shows the zygomatic arch at the inferior region, the corresponding location of the transducer should be located quite lower than Figure 3B.
Please choose the nomenclature of the probe and transducer in the text and table.
Overall, the contents of the manuscript are interest, however, this seems like a chapter of a book rather than a review paper.
Author Response
Reviewer 3
Comment
Dear authors,
The paper is interesting and well-illustrated. However, I have a few comments as follows:
Figure 1 contains incorrect information. As the first figure of this manuscript delivers key anatomical information, I suggest that the Figure 1 be revised and the nomenclature be correctly used when indicating the facial muscles. For example, zygomatic major and minor should be revised as zygomaticus major and minor muscles.
What was indicated as the buccinator is actually the risorius muscle. The location of the temporalis muscle is also incorrect. Please carefully recheck the nomenclature and the location of the all the facial muscles before further submission. The superficial aponeurosis of the masseter muscle should be drawn longer than how it is depicted in the manuscript. Please refer to previous anatomical research papers that describe the morphology of the masseter muscle.
Response:
We appreciate the kind comment from the reviewer.
First, we have revised the incorrect nomenclature. The zygomatic major and minor muscles have been revised as zygomaticus major and minor muscles.
Second, the incorrect labeling regarding the buccinator and risorius muscles in Figure 1 has been corrected.
Third, regarding the location of the temporalis muscle, we have moved the muscle bulk more anteroinferiorly. The distal aponeurosis has been extended more inferiorly until it submerges underneath the masseter. The dashed line has been used to show the imaginary position of the zygomatic arch.
Fourth, the superficial aponeurosis of the masseter has been redrawn to make it longer.
Comment:
The temporal fossa indicated in Figure 3A should be revised. The temporalis muscle should be found in the temporal fossa as a hypoechoic structure in the ultrasonography image. However, the authors did not indicate the temporalis muscle in the temporal fossa. When the temporal fossa is observed in the ultrasonography image, the temporal bone surface can be found deep to the temporalis muscle as a hyperechoic structure.
The location of the transducer, ultrasonography image and corresponding illustration are incorrect in Figure 3. As the ultrasonography image shows the zygomatic arch at the inferior region, the corresponding location of the transducer should be located quite lower than Figure 3B.
Response:
We appreciate the kind comment from the reviewer.
First, the temporalis muscle has been labelled in revised Figure 3A.
Second, the term “temporal fossa” has been replaced by “temporal bone” in accordance to the reviewer’s comment.
Third, the subgraph for showing the transducer’s position has been revised by relocating the footprint of the transducer more inferiorly.
Comment:
Please choose the nomenclature of the probe and transducer in the text and table.
Response:
We have unified the nomenclature as “transducer” across the whole manuscript.
Comment:
Overall, the contents of the manuscript are interest, however, this seems like a chapter of a book rather than a review paper.
Response:
We appreciate the kind comment of the reviewer. The article is the combination of the pictorial essay and narrative review, which serve as a useful guideline for ultrasound guidance botulinum toxin injections for facial muscles. In the section of “literature search”, we also demonstrate our systematic method to identify relevant studies to let our article more like a review.
Reviewer 4 Report
This review primarily focused on superficial facial anatomy, ultrasound imaging, and scanning techniques. While the title includes the guidance for botulinum toxin injections, it only includes some cases that botulinum neurotoxin injections can be used for the intervention and did not provide or review any guidance on how to use the US for botulinum neurotoxin injections. Most cases authors referred don't involve the US-guided injections. Indeed, even section 6 on the literature review only focused on the how-to US to imagine the facial muscles but did not include the review on using US as a guide for botulinum neurotoxin injection. Indeed, US has been used as a guide for botulinum injections for numeral diseases, including the facial muscles, such as bruxism. Therefore, this review as current form does not provide much relevance to clinical uses of botulinum neurotoxin.
Author Response
Reviewer 4
This review primarily focused on superficial facial anatomy, ultrasound imaging, and scanning techniques. While the title includes the guidance for botulinum toxin injections, it only includes some cases that botulinum neurotoxin injections can be used for the intervention and did not provide or review any guidance on how to use the US for botulinum neurotoxin injections. Most cases authors referred don't involve the US-guided injections. Indeed, even section 6 on the literature review only focused on the how-to US to imagine the facial muscles but did not include the review on using US as a guide for botulinum neurotoxin injection. Indeed, US has been used as a guide for botulinum injections for numeral diseases, including the facial muscles, such as bruxism. Therefore, this review as current form does not provide much relevance to clinical uses of botulinum neurotoxin.
Response:
We appreciate the kind comment from the reviewer. Like what the reviewer mentioned, there has been guidelines of using ultrasound to guide injection of spasticity and bruxism. However, as the facial muscles are superficially located, some cosmetologists may think it to be unnecessary to use ultrasound guidance for injecting facial muscles. That is why in the section of “FUTURE PERSPECTIVES AND LIMITATIONS”, we would acknowledge “Although the botulinum toxin is widely used for rejuvenation and in cosmetic dermatology, few studies have investigated the role of US guidance in comparison with landmark-based injections” (line 541-543). We also emphasize that “future studies are warranted to conclusively establish the clinical effectiveness and safety of such interventions” (line 546-547).
Furthermore, in accordance to the review’s comments, we have revised our title as “Ultrasound Imaging of the Facial Muscles and Relevance with Botulinum Toxin Injections: A Pictorial Essay and Narrative Review”. In this article, we would not discuss much about the effect of guided botulinum toxin injection on the cosmetic issues as the number of available studies is very limited. I think it would address the concern of the reviewer regarding why we focus on ultrasound imaging and scanning techniques.
Lastly, the limitation of ultrasound guided botulinum toxin injections on facial muscles as well as the necessity of a guideline for ultrasound guidance have been added as “Several limitations of using US guidance for injecting facial muscles should be acknowledged. First, utilization of US guidance for most facial muscles may be excessive since the majority of facial muscles can be identified with anatomical surface landmarks, such as most movement disorders (hemifacial spasm, blepharospasm and other forms of facial dystonia). Second, the localization of the target muscles by US imaging is challenging to validate. The cosmetologists may consider to use US imaging while injecting muscles besides some vital neurovascular structures, such as the facial artery and facial nerves. A guideline made by the consensus of the cosmetologists is needed in the future as to what circumstance would the use of US imaging be necessary because most cosmetologists do not utilize the technique currently” (line 548-557).
Round 2
Reviewer 3 Report
N/A
Reviewer 4 Report
The revisions are fine.